# Serial Scammers and Attack of the Clones: How Scammers Coordinate Multiple Rug Pulls on Decentralized Exchanges

## ABSTRACT

We explored in this work the ubiquitous phenomenon of *serial scammers*, who deploy thousands of addresses to conduct a series of *similar* Rug Pulls on popular decentralized exchanges (DEXs). We first constructed a list of about 163,000 scammer addresses behind all 1-day Rug Pulls on the two most popular DEXs, Uniswap (Ethereum) and Pancakeswap (BSC), and identified many distinctive *scam patterns* including *star-shaped*, *chain-shaped*, and *majority-flow* scam clusters. We then proposed an algorithm to build a complete *scam network* from given scammer addresses, which consists of not only scammer addresses but also supporting addresses including depositors, withdrawers, transferrers, coordinators, and most importantly, *wash traders*. We note that profit estimations in existing works on Rug Pulls failed to capture the cost of *wash trading*, leading to inflated figures. Knowing who the wash traders are, we established a more accurate estimate for the *true profit* of individual scam pools as well as of the *entire* (serial) scam network by taking into account the wash-trading expenses.

**ACM Reference Format:**

. 2024. Serial Scammers and Attack of the Clones: How Scammers Coordinate Multiple Rug Pulls on Decentralized Exchanges. In *Proceedings of The Web Conference (WWW '25)*. ACM, New York, NY, USA, 14 pages. https://doi.org/XXXXXXX.XXXXXXX

## 1 INTRODUCTION

The total crypto scam revenue from 2019 to 2023, according to the latest 2024 Crypto Crime Report by the leading blockchain analytics firm Chanalysis [8, p. 104], reached a staggering amount of nearly US $40 billion. The report also shows that *Rug Pull*, a common type of scam in the decentralized finance (DeFi) ecosystem, was among the top three fastest growing scams in 2023 [8, p. 105]. Rug Pull, first reported in 2021 [7, 38], refers to scams in which the developer(s) of a cryptocurrency project (usually a new token) suddenly vanished with investors' fund, leaving their purchased assets worthless. Rug-Pull scams were responsible for the loss of more than US $100 million in 2023 alone according to Immunefi's Crypto Loss Report [16], and are still costing millions of dollars every month in 2024[1]. Immunefi's reports [16] also identified Etherem and BNB as the two most targeted chains by hacks and Rug Pulls in 2023-2024.

Whenever there are scammers, there must be *serial scammers*. Some evidence of that was initially observed in Xia *et al.* [38] when

[1]Note that Immunefi's report considers only Rug Pulls for its "fraud" category.

they expanded their dataset of scam tokens on Uniswap by including also tokens that were created by known scammer addresses, which were later manually confirmed to be scam tokens (see [38, Sect. 4.4]). In other words, there are addresses that created multiple scam tokens on Uniswap. Recently, in a comprehensive study of the token ecosystem in Ethereum and Binance Smart Chain (BNB), Cernera *et al.* [6] also discovered a number of scammer addresses that performed *multiple* 1-day Rug Pulls (exchange pools that were rugged within a day). However, both works assumed that scammer addresses are independent, and the Rug Pulls carried out by them are unrelated, leaving the case of single scammers coordinating multiple scam addresses for future research (see [6, Sect. 11]).

Xia *et al.* [38] also investigated a related concept of *collusion* addresses of a scam token/pool creator, which are addresses that likely belong to the same scammer (there were money flow between them and the main scammer address who created the token and the pool) and operate on *the same scam pool*. While collusion addresses form a small part of our study, they are defined for *individual* scam pools and do not capture the setting of serial scammers where *multiple* addresses operate on *multiple* related scam pools. Xia *et al.* also noted that there might be more complex networks (e.g. to launder their fund) operating behind the scams that their analysis failed to capture, and hence there might be many more scam addresses not identified by their heuristics (see [38, Sect. 6.3]). Similar to [38] and [6], other existing works on Rug Pulls [15, 21, 23, 40] only investigated snapshots of the Rug-Pull scam landscape by zooming in to either individual scam tokens/pools or individual scammer addresses and treating them as independent entities.

In this paper, we seek to address the aforementioned research gap and settle the open problems raised by Xia *et al.* [38] and Cernera *et al.* [6]. To that end, we go one step deeper into the world of sophisticated serial (Rug-Pull) scammers by investigating groups of tightly connected addresses (supposedly belonging to the same scammers or scam organizations) that were behind multiple Rug-Pull scam tokens (ERC-20 on Uniswap and BEP-20 on Pancakeswap), which, not surprisingly, have highly similar contracts. In other words, we zoomed out and connected the dots to reconstruct a more comprehensive and accurate picture of how serial Rug-Pull scammers organized their operations.

To facilitate the exposition of the serial scammers, we restricted our investigation to *1-day Simple Rug Pull* tokens, which are easy to identify and prevalent on Ethereum and BSC. Such tokens lived for only one day and were paired with a high-value token such as ETH or BNB in a pool, the liquidity of which was provided and removed by a scammer address within a day (see [6]). As shown in [6], 1-day Rug-Pull tokens were abundant on Ethereum and BSC, accounting for nearly 50% of all tokens[2] on these chains from inception to March 2022. Note that 1-day Simple Rug Pulls are much easier to detect with higher confidence compared to longer-life Rug

[2]It was reported in [6] that approximately 60% of tokens on Ethereum and BSC Smart Chain (BNB) have lifetime shorter than a day, and more than 80% among which were Rug Pulls at their time of study.

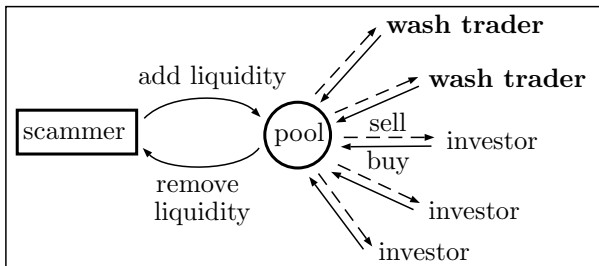

**Figure 1:** Typical activities in a DEX scam pool. There can be one or more *scammer* addresses behind each pool. *Wash traders* bought small amounts of scam tokens to increase its price and generate fake activities. The arrow refers to the flow of ETH/BNB. If the scam token is a Trapdoor [15] then it's possible to buy but impossible to sell the scam token to obtain ETH/BNB (henced the dashed arrows).

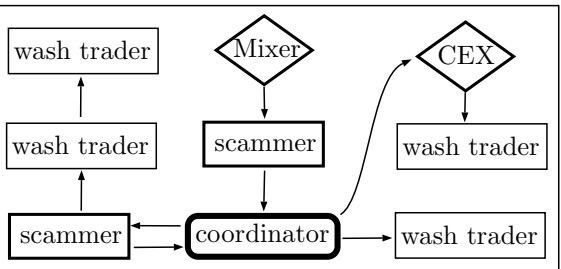

**Figure 2:** Typical transactions within a *scam network*. A *coordinator* address often funds several scammer addresses and sometimes *wash-trader* addresses. It also sometimes plays the role of a *depositor* and transfers large amount of fund to CEX/mixer. Wash traders can be funded by any node in the network. Sometimes they obtain fund directly from a CEX/mixer, playing the role of a *withdrawer*.

Pulls, which were usually labeled by less straightforward rules, e.g. inactive tokens (no activities for more than a month) where the corresponding pools had their liquidity completely removed or had their price dropped more than 90% at some point. Longer-life Rug-Pull tokens could be potentially mixed up with low-performing tokens where the creator decided to remove the liquidity after a long period of no profit without any ill intent.

*Wash trading* activities in the cryptocurrency ecosystem have been observed and studied in different contexts. For example, wash trading can be carried out to artificially increase the trading volume of cryptocurrency exchanges [28, 33] to influence the perception of their popularity. Wash trading can also be used to boost the trading volume of a Non-Fungible Token (NFT) to reap the reward from an NFT marketplace or inflate its price for reselling [9, 22, 34]. Wash trading activities were also observed in the context of Rug Pulls of ERC-20 tokens on Uniswap in Xia *et al.* [38, Sect. 5.3.3], under their investigation of collusion addresses. However, under their heuristics, only wash traders that have a *direct* transaction with the scammers, i.e. 1-hop neighbors, are included. Even the recently developed *A-A Wash-Trading Detector* for Uniswap V2 on Dune from SolidusLab [18, 19] can only detect self wash trading (or 0-hop wash trader). This simple wash-trading model fails to capture more sophisticated scams in which wash traders are multiple hops away, as frequently observed from our datasets.

In our work, by building the *scam networks* of scammer addresses and scam-supporting addresses of serial scammers, we were able to identify all *wash-trader* addresses, which were funded by the scam networks and bought scam tokens to pump up their prices in order to lure the real investors in (see Fig. 1 and Fig. 2). This allows us to estimate more accurately the *real profits* of the scammers. As an example, the creator of the Uniswap pool that pairs ETH and a scam token called PUMPKIN added 1.8 ETH and removed 9.27 ETH, seemingly reaped a profit of more than 7 ETH (>$10,000) after mere 37 minutes, creating the illusion of a highly successful scam. However, as discussed in Section 6.2, it turns out that all major investors were wash traders, and the *real profit* for the creator of PUMPKIN, after deducting the wash-trading expense, is almost zero! Our contributions are summarized below.

- We constructed large 1-day Rug-Pull datasets on Unisw - ap [32] and Pancakeswap [26] V2, consisting of nearly 200,000 scam pools and 163,000 scammer addresses behind them.
- We formally defined and identified numerous *scam patterns* in our datasets, including the *star-shaped*, *chain-shaped*, and *majority-flow clusters*. The longest scam chain and the largest scam star consist of 274 and 585 scammer addresses, respectively. These patterns revealed distinctive ways serial scammers coordinated multiple scams on DEXs.
- We further grouped the scammer addresses that interact with each other via a direct ETH/BNB transfer or via a scam pool into scam clusters, and found that token contracts used within each cluster are mostly similar, which indicates that they could be clones from the same source (same scammer).
- Using a novel network explorer algorithm, we were able to reconstruct the complete picture of how serial scammers actually work. In particular, we propose a *network-aware* profit formula that factors in the wash-trading expenses to achieve a more accurate scam profit estimate.

## 2 BACKGROUND

### 2.1 Ethereum and BNB Smart Chain

**Ethereum** is the second-most popular blockchain after Bitcoin, with a market capitalization of over US$300 billion at the time of this study [12]. At the time of its launch in 2015, Ethereum had drawn great attention to the blockchain community by introducing Ethereum Virtual Machine (EVM) and a smart contract concept, becoming a pioneer in contract-supporting blockchains. Smart contracts are executable pieces of code holding business logic. Many applications have been implemented using smart contracts, boosting blockchain to rapidly develop and be widely applied to different domains, such as supply chain, health care, and governance [5, 17, 24]. One of the well-known smart-contract-based applications is fungible tokens.

**Fungible tokens** are the financial applications used widely as digital assets such as company shares, online game assets, or fiat currencies. Fungible tokens (ERC-20 on Ethereum or BEP-20 on BSC) must follow the common standard [2, 14] by implementing a

set of functions and events (see Table 5 in Appendix A). **Accounts** are the basic unit in blockchain that is represented by a 20-byte length unique address. Ethereum and BSC accounts are classified into externally owned account (EOA) and contract account (CA). The former is controlled by users via a private-public key pair mapped to the address, while the latter is managed by a contract that contains executable code. **Transactions** in blockchains are messages between two accounts. Transactions record all activities on blockchains, such as deploying a new smart contract or transferring a digital asset. A fee will be charged to a user when creating a transaction to pay the miners. Transactions are classified based on the type of the sender: a normal transaction is sent from an EOA, while an internal transaction is sent from a CA.

**BNB Smart Chain (BSC)**, a hard fork of Ethereum, was established in 2020 with new features to boost the performance. One of the important updates is the use of a different consensus, allowing faster transaction processing times with lower fees. Similar to Ethereum, BSC is also a smart-contract-supporting blockchain with a very similar token standard (BEP-20).

## 2.2 Uniswap and Pancakeswap

Decentralised exchanges (DEXs) are financial platforms that operate based on price determination mechanisms such as order books and automated market makers (AMM), allowing users to exchange their digital assets without the involvement of central authorities [36]. Uniswap [32] is one of the top popular DEXs, which debuted on Ethereum in 2018. Uniswap is the first DEX that adopted the AMM mechanism successfully with the concept of exchange pools and liquidity providers. An exchange pool in Uniswap operates like a money-exchange counter of two arbitrary currencies (fungible tokens). Exchange liquidity in a pool is provided by one or multiple users (liquidity providers). Anytime a provider adds liquidity (in the form of two corresponding tokens) into a pool, the pool's smart contract will "mint" LP tokens and send them back to a provider as liquidity shares. A provider can send LP tokens to the pool to withdraw their funds anytime they need. When a pool receives LP tokens from a provider, its smart contract will "burn" them and send the fund back to a provider.

Although Uniswap has launched its fourth version, other versions are still operating as independent platforms. Among them, Uniswap version 2 (UniswapV2) entirely outperforms other versions in terms of the number of listed tokens and pools [11]. Due to the popularity of this version and its open-source smart contracts, more than 650 DEXs across different blockchains are the forks of UniswapV2 [13], and PancakeswapV2 [26] is the most successful fork on BSC. As such, we choose to study UniswapV2 and PancakeswapV2, noting that our approach is also applicable to other forks of UniswapV2 and other similar DEXs.

## 3 ONE-DAY RUG PULL DATASETS

We first define 1-day Rug Pull scam and then discuss how we constructed the datasets of scam pools, scam tokens, and scammer addresses on Uniswap V2 (Ethereum) and Pancakeswap V2 (BSC).

## 3.1 One-Day Rug Pull

We follow the definition of 1-Day Rug Pull in Cernera *et al.* [6, Sec. 7.1]. The definition of scammer addresses were first discussed in Xia *et al.* [38, Sec. 5.3]. See Fig. 7 for an example of the transaction history of a typical Rug-Pull scammer address.

*Definition 3.1 (1-Day Rug Pull).* A *1-day exchange pool* is a pool that pairs a higher-value (lower reserve) token and a lower-value (higher reserve) token in which the first and last events happen within a day. An exchange pool is called a *1-day simple Rug Pull* if a) it is 1-day, b) the lower-value token is paired in this pool only, and c) it has one mint event and one burn event that burns at least 99% of the minted LP tokens. The corresponding lower-value token of a 1-day simple Rug-Pull pool is also called a 1-day simple Rug-Pull token. We also refer to them as *scam pool* and *scam token* for short.

*Definition 3.2 (Scammer Addresses).* Given a cam pool, we refer to the addresses of the scam token creator, the scam pool creator, the liquidity provider, and the liquidity remover as *scammer addresses* behind/associated with the pool.

## 3.2 Data Collection

We collected data on Uniswap and Pancakeswap from the time they were launched to the time of this study (July 2024).

**Exchange Pools.** Uniswap works based on three main contracts: `Factory`, `Pair`, and `Router`. The `Factory` generates an *exchange pool* (a `Pair`) for users from two given tokens. The `Factory` stores the addresses of created `Pairs` while a `Pair` stores the addresses of two listed tokens. We first used Web3.py [35] to query all created pools by calling the `allPairs` function of the `Factory`. For each collected pool, we then called two functions `token0` and `token1` from its contract to retrieve the pair of listed tokens. Finally, we used Etherescan APIs [31] and BSCscan APIs [30] for retrieving related information of collected pools and tokens, including their creator address and their contract source codes. As a result, we collected 356,295 pools and 343,637 tokens on Uniswap, and 1,694,058 pools and 1,510,774 tokens on Pancakeswap.

**Pool Events.** Pool events are the logs of a `Pair` written down when the state of any property changes. In our study, we collected four pool events, including `Mint`, `Burn`, `Transfer`, and `Swap`. The first three events occur every time LP-token information is changed. For example, an exchange pool emits a `Mint` event each time LP-tokens are minted for a new liquidity adding. Similarly, a `Burn` event is raised when a pool burns the LP tokens received from a liquidity provider. A `Transfer` event is recorded any time an LP token is transferred from one address to another (ownership change). Unlike these three events, a `Swap` event is written each time a user swaps tokens in an exchange pool. To collect these events, we used the "getLogs" APIs from Etherscan and BSCscan. To reduce time and computation cost, we only download the first 1000 events of each pool, as it also will not impact our goal of collecting 1-day scam pools/tokens. The downloaded data was then decoded to extract useful information using our event decoder. In consequence, we gathered 2.2 million `Mint` events, 1.0 million `Burn` events, 4.7 million `Transfer` events and 49.9 million `Swap` events on Uniswap (resp. 17.3 million, 5.1 million, 38.0 million, and 154.9 million events on Pancakeswap).

**Rug Pulls and Scammers.** Next, with the pool events we gathered in the previous steps, we identify Rug Pull scams on each DEX by using the one-day Rug Pull definition. Notably, we collect all pools that only fire one `Mint` and one `Burn` event and examine if

99% of liquidity is burned within a day of it being added. Moreover, we only focus on ETH pools on Uniswap and BNB pools on BSC. This allows us to accurately assess the costs and benefits associated with these scams. According to our analysis, 96% number of pools on Uniswap are ETH pools, and 88% number of pools on Pancakeswap are BNB pools. Thus, the missing cases are not important for our study, but it is difficult to estimate the cost from the token prices. Finally, we extract scammers from identified rug pulls. It is worth noting that we exclude all public addresses (e.g., CEX, bots, bridges) and old addresses that have no transactions within our data collection period. These old addresses appear in our dataset because their tokens were created before the platforms were launched. Ultimately, 161,339 (47%) scam pools and 145,654 unique scammers are determined on Uniswap, and 32,336 (2%) scam pools and 18,346 unique scammers are found on Pancakeswap.

# 4 SERIAL SCAM PATTERNS: DETECTION AND ANALYSIS

We explore in this section several distinctive *funding patterns* that reveal how scammer addresses, which potentially belong to the same serial scammers, receive fund to carry out their Rug-Pull scams and transfer the scammed money to another address in a *highly coordinated* manner. For one of our funding patterns: *simple* scam chain, we found the top 50 highest average transfer amounts per chain to average 902 ETH for Uniswap and 138 BNB for Pancakeswap. This is the first step in our investigation of serial scammers on DEXs where we deviate from most existing approaches [6, 15, 21, 23, 38, 40], which often treat different scams as unrelated ones.

We henceforth define the following concepts to facilitate our discussion. Given an address A, an *in-transaction* is a transaction from another address B to A, and B is called an *in-neighbor* of A. Similarly, an *out-transaction* is a transaction from A to another address C, which is referred to as an *out-neighbor* of A. We refer to the buys and sells of a scam token as *swap-ins* (paying ETH/BNB to the pool) and *swap-outs* (geting ETH/BNB from the pool).

## 4.1 Star-Shaped Scam Patterns

We start our scam pattern exploration by defining three types of scam star, representing a commonly found pattern in which scammer addresses are coordinated by a center address (coordinator).

*Definition 4.1 (Scam Star).* A *scam star* consists of a *center* address $c$ (coordinator) and $n \geq 5$ *scammer* addresses $s_1, \ldots, s_n$ (satellites) that satisfy one of the following patterns.

- **OUT-star (common funder)**: the satellites $s_1, \ldots, s_n$ received at least 100% of the cost to create their first scam from the center $c$ but sent no fund to the center. The corresponding in-transaction from $c$ must be the largest in-transaction each satellite received *before* conducting the first scam.
- **IN-star (common beneficiary)**: the satellites $s_1, \ldots, s_n$ received no fund from the center $c$, but transferred at least 90% of their last scam revenue to the center. The corresponding out-transaction to $c$ must be the largest out-transaction from each scammer *after* conducting their last scam.
- **IN/OUT-star (common funder/beneficiary)**: the satellites $s_1, \ldots, s_n$ received at least 100% of the cost to create

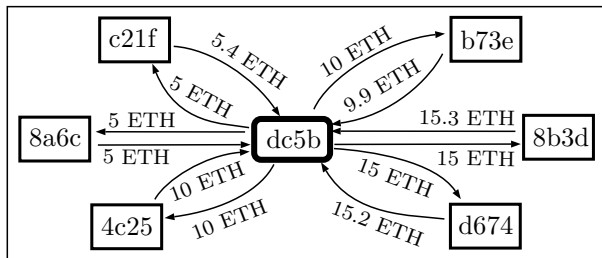

**Figure 3:** Examples of an IN/OUT-star with **dc5b** as the center/coordinator[3]. The six satellites are all scammer addresses.

their first scam from the center $c$. The corresponding in-transaction from $c$ must be the largest in-transaction each satellite received *before* conducting the first scam. Moreover, the satellites transferred at least 90% of their last scam's revenue to the center. The corresponding out-transaction to the center must also be the largest out-transaction from each scammer *after* conducting the last scam.

Examples of an IN-star and an IN/OUT-star are given in Fig. 3. Note that while we require that the funding amount from the center must cover 100% of the scam cost (in OUT-stars and an IN/OUT-stars), the value of the OUT transaction only needs to cover at least 90% of the last scam's revenue. This reflects our observation that a scammer address may also operate as a wash trader, who spends some small portion of the sum received from the funder and from its scam pool to buy (i.e. *wash trade*) scam tokens from scam pools created by *other* scammer addresses. More details on how to detect stars are left to Appendix C. We ran our star detection algorithm on the entire Uniswap/BSC datasets and report the statistics below.

| Type | #Stars | Size | Fund In | Fund Out | Period (days) | #Scams |
|---|---|---|---|---|---|---|
| IN | 1575 | 585, 19 | 1173, 6 | | 925, 95 | 585, 19 |
| OUT | 61 | 66, 9 | | 1310, 42 | 475, 56 | 66, 10 |
| IN/OUT | 73 | 159, 15 | 1247, 37 | 1263, 36 | 475, 55 | 159, 16 |

**Table 1:** Statistics for *scam stars* found in our **Uniswap** scammer dataset. Maximum and average values are rounded and reported, e.g., the maximum and average size of an IN scam star is 589 (satellites) and 19, respectively.

| Type | #Stars | Size | Fund In | Fund Out | Period (days) | #Scams |
|---|---|---|---|---|---|---|
| IN | 64 | 228, 20 | 24,4 | | 758, 187 | 230, 22 |
| OUT | 0 | | | | | |
| IN/OUT | 7 | 12, 8 | 1, 1 | 2, 1 | 17, 4 | 12, 8 |

**Table 2:** Statistics for *scam stars* found in our **BSC** scammer dataset. Maximum and average values (in BNB) are rounded and reported.

## 4.2 Max-In-Max-Out Scam Chain

*Definition 4.2 (Simple Scam Chain).* A *simple scam chain*, also referred to as a *max-in-max-out* scam chain, is a list of $n$ scammer addresses $s_1, s_2, \ldots, s_n$ ($n \geq 2$) satisfying the following conditions.

- (C1) $s_i$ is the largest funder of $s_{i+1}$, and $s_{i+1}$ is the largest beneficiary of $s_i$, for every $i \in \{1, \ldots, n-1\}$.

---

[3]The center's full address is 0xbfc6cc4676aef7216e597d45d68463097520dc5b.

- (C2) The transfer(s) from $s_i$ to $s_{i+1}$ occurred after $s_i$ has completed its last scam and before $s_{i+1}$ started its first scam.

A max-in-max-out scam chain is called *maximal* if no other EOAs can be added to the chain to obtain a longer chain. We only consider maximal scam chains in this work. It is obvious that each scammer belongs to at most one max-in-max-out chain.

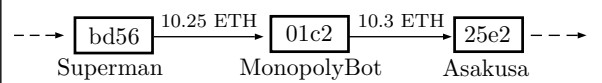

**Figure 4:** An example of three consecutive scammer (partial) addresses[4] as part of a (maximal) *simple scam chain* of length 47. Each address performed one Rug Pull (token names are given), and then transferred some fund to the next one.

| DEX | #Chains | Length | Ave. Transfer | Period (days) | #Scams |
|---|---|---|---|---|---|
| Uniswap | 4494 | 274[5], 4 | 642, 35 | 369, 3 | 339, 6 |
| Pancakeswap | 519 | 19, 2 | 817, 19 | 61, 3 | 125, 5 |

**Table 3:** Statistics for *scam chains* found in our Uniswap and Pancakeswap scammer datasets. Maximum and average values (in ETH/BNB) are rounded and reported.

A simple algorithm can be designed to find all (maximal) max-in-max-out scam chains among a given list of scammer addresses. Our findings on both datasets are reported in Table 3. Except the number of chains, the rest are measurements per chain.

### 4.3 Majority-Flow Cluster

While the max-in-max-out scam chains capture many cases where each scammer address has a major funder and then sent most scammed fund to another scammer address, they miss the case where there are *more than one* major funders and/or beneficiaries. For example, when two addresses funded the scammer with 50 and 51 ETH, only the one funding 51 ETH will be recorded by the chain.

We propose below the more sophisticated concept of *majority-flow cluster*, which takes into consideration the funding amount required to fund a scam (when providing liquidity) and its revenue (when removing liquidity). Such a cluster consists of three kinds of (scammer) addresses: *input*, *internal*, and *output*. Each *internal* address received 100% funding from input addresses and/or other internal addresses (called the *major funders*) to carry out its first scams, then transferred at least 90% of its last scam's revenue to other internal and/or output addresses (called the *major beneficiaries*). Note that a scammer address may also spend some of its scam revenue on wash trading other scam pools (hence only 90% is required instead of 100% to account for those small spendings). *Input* addresses and *output* addresses behaved similarly, but the funders of input addresses and beneficiaries of output addresses do not belong to the cluster, respectively.

The majority-flow cluster concept reflects more accurately the major flow of funding for scams between scammer addresses on a DEX. An example of a partial majority-flow cluster is given in

Fig. 8 (Appendix D). Note that the max-in-max-out chain can only capture scammer addresses along the path -9ac2-29cc-d378-c80f-2970- instead of the whole cluster.

*Definition 4.3 (Majority-Flow Cluster).* Given a set of scammer addresses $S$, a *majority-flow cluster* is subset $C$ of $S$, $|C| \geq 2$, where every $s \in C$ has a set of in-neighbors $F(s)$ called the *major funders* and/or a set of out-neighbors $B(s)$ called the *major beneficiaries* satisfying the following properties.

- (P1) The set $F(s) \subset S$ consists of the *minimum* number of in-neighbors with largest in-transaction values (top funders) *before* the *first* scam of $s$ occurred that together provide enough funding for $s$ to fund its first scam.
- (P2) The set $B(s) \subset S$ consists of the *minimum* number of out-neighbors with largest out-transaction values (top beneficiaries) *after* the *last* scam of $s$ occurred that together cover at least 90% of the revenue of the last scam of $s$.
- (P3) For every $s \in C$, it holds that $F(s) \subseteq C$ or $F(s) \cap C = \varnothing$, and $B(s) \subseteq C$ or $B(s) \cap C = \varnothing$. Moreover, either $F(s)$ or $B(s)$ or both must be a subset of $C$.
- (P4) For every $s, s' \in C$, it holds true that $s' \in F(s)$ if and only if $s \in B(s')$.
- (P5) $C$ is *connected* in the sense that for every $s, s' \in C, s \neq s'$, there is a path $s = v_1, v_2, \ldots, v_k = s'$ where $v_{i+1} \in F(v_i)$ or $v_{i+1} \in B(v_i)$, for every $i = 1, \ldots, k-1$.

Similar to the chains and stars, we only consider *maximal* majority-flow clusters, i.e. no other EOAs can be added to obtain a larger one. Each scammer belongs to at most one cluster (see Theorem D.1).

Although the majority-flow clusters require very strict properties, we were still able to find a good number of them on both chains (see Table 4). Those with *width* two (see Appendix D) have a chain shape (but not the same as the simple chains), whereas those with width larger than two represent the more sophisticated clusters with nodes having more than one major funder/beneficiary.

| DEX | No. Clusters | Size | Width | Fund In | Fund Out |
|---|---|---|---|---|---|
| Uniswap | 5298 | 156, 4 | 7, 2 | 1644.8, 31.8 | 1797, 33 |
| Pancakeswap | 817 | 19, 2 | 3, 2 | 816.8, 14.5 | 816.8, 15 |

**Table 4:** Statistics for *majority-flow clusters* found in our Uniswap and Pancakeswap scammer datasets. Maximum and average values are rounded and recorded.

## 5 SCAM CLUSTERS ANALYSIS

We have seen in Section 4 how scammer addresses fund themselves via scam clusters with special shapes. In this section, we investigate more general *scam clusters*, each of which consists of scammer addresses linked together via a direct ETH/BNB transaction or a scam pool. Note that all scam patterns investigated in Section 4 make use of direct ETH/BNB transactions only.

To corroborate our view that all such scammer addresses in the same cluster likely belong to the same serial scammer, we developed *AST-Jaccard score* (see Appendix E), a new code similarity score based on a careful integration of contract source code's *abstract syntax tree* (AST), *hash function*, and *Jaccard similarity* to overcome typical code obfuscation techniques used in contract cloning. We measured the *similarity* among a large number of scam contracts,

---

[4]The first scammer address is 0xC3E8290045952D520f4c2Eb7E8725CaBc4c8B5D6.
[5]The first address is 0x79daa9236e6825f023ab3ebd2cecfe94b48789d1.

and found that *intra-cluster* contracts are mostly similar (average similarity score greater than 0.7) while *inter-cluster* contracts are mostly dissimilar (average similarity score less than 0.3). This is yet another strong indicator that *scammer addresses within each cluster are controlled by the same serial scammer*, apart from the fact that such addresses are already either behind the same scam pools or tightly connected in the transaction network. With the new concept of scam clusters and the affirmative contract similarity analysis of them, we have further broadened our understanding of how serial scammers created similar contracts to run a series of Rug-Pull scams on popular DEXs.

## 5.1 Generating Scam Clusters

*Definition 5.1 (Scam Cluster).* Given a dataset $S$ of all scammer addresses, a *scam cluster* is a connected component (see below) $C = (V(C), E(C))$, where $V(C) \subseteq S$ is a set of *scammer* addresses and $E(C) \subseteq V(C) \times V(C)$ is a set of edges among them, satisfying the following conditions.

- (C1) An edge $e = (u, v) \in E(C)$ exists if and only if $u$ and $v$ had a direct ETH/BNB transaction on the blockchain, or $u$ and $v$ are different scammer addresses associated with a common scam pool.
- (C2) $C$ is connected, that is, for every $u, v \in V(C)$, $u \neq v$, there exists a path from $u$ to $v$ in $C$.

Note that we only consider *maximal* scam clusters, i.e., no new scammer address can be added to achieve a larger one.

Given a dataset $S$ of all scammer addresses, one can identify all scam clusters by first forming an undirected graph $G = (V, E)$ with $V = S$ and $E$ is formed using (C1), replacing $V(C)$ and $E(C)$ by $V$ and $E$, respectively, and then find all of its connected components. The scammer addresses belonging to each connected component form a scam cluster. We ran this simple algorithm on both the Uniswap and BSC datasets of scammer addresses and identified *all* (1-day simple Rug Pull) scam clusters on these DEXs. The results show that 109,471 groups on Uniswap and 14,561 groups on Pankaceswap are formed from 145,654 scammers and 18,346 scammers, respectively. The biggest Uniswap cluster contains 4,155 unique scammers, while the biggest cluster on Pancakeswap is established by 571 different addresses. There are also many one-scammer groups on both platforms that occupy nearly 92% Uniswap clusters and 88% Pancakeswap clusters.

## 5.2 Scam Cluster Analysis

We collected available token contracts from all the scammers in the scam clusters in Section 5, and obtained 111,300 token contracts for 74,254 clusters on Uniswap and 14,147 token contracts for 9,610 clusters on Pancakeswap. We first examined the similarity among contracts that were used by individual scammer addresses and found that nearly 68% of scammer addresses on Uniswap and 61% of scammer addresses on Pancakeswap deployed multiple contracts with over 80% similarity. Moreover, 2,038 Uniswap scammer addresses and 411 Pancakeswap scammer addresses reused the same contract (100% similarity) repeatedly. Among them, **e937**[6] on

Uniswap and **acd0**[7] on Pancakeswap deployed the same contract at most 150 times and 24 times, respectively.

**Intra-cluster similarity.** Fig 9 (Appendix E) shows the statistic of intra-cluster similarity with different sizes of groups on Uniswap. Specifically, 5,483 groups (33%) on Uniswap have a similarity of over 95%. This proportion does not much change for the set of large-size groups. The opposite phenomenon is observed for the distribution of similarities on Pancakeswap. The proportion of over 95% similarity groups is higher than the proportion on Uniswap with the value of 37% (see Fig 10, Appendix E). However, this number decreases by 12% in the large-size group. There are only a few groups set larger than 50 so we do not do a statistic for these groups. The average intra-cluster similarities among all groups are 74% and 79% for Uniswap and Pancakeswap, respectively.

**Inter-cluster similarity.** Due to high computational cost, in our calculation, we randomly select up to 100 tokens in each group to compare with those in other groups. We also select randomly 500 groups to pair with the chosen group. Then we repeat the calculation 10 times and the final result is obtained by taking the average. The results in Fig 11 and Fig 12 (Appendix E) indicate that tokens in one group are dissimilar to tokens in other groups. Regardless of group size, the similarity is always below 39% for Uniswap clusters and 30% for Pancakeswap clusters. The average similarity for any pair of clusters on Uniswap is 27%, while that on Pancakeswap is 20%.

## 6 SCAM NETWORKS AND PROFIT ESTIMATES

We are now ready to present the most complete picture of how a Rug-Pull *scam network* (belonging to a serial scammer/scam organization) operates on Uniswap and Pancakeswap. A scam network contains not only *scammer* addresses but also other associated addresses that serve distinctive roles in the overall scam operation including *wash traders, transferrers, depositors, withdrawers,* and *coordinators*. A scam network contains the scam chains and scam stars (Section 4), as well as scam clusters (Section 5) as its subgraphs.

Apart from helping us obtain a complete view of the entire operations of a serial scammer, a concrete benefit of having a complete scam network reconstructed is that the knowledge of the wash traders in the network allows a more accurate computation of the true scam profits. Some of the existing research in DEX Rug Pull were aware of the wash trading activities [6, 15, 29, 38] but leave it as a future work. This is because without the reconstruction of a scam network, it is impossible to calculate the *wash-trading component*. Our work seeks to address this research gap.

## 6.1 Node Labelling

We first define the main roles performed by addresses in a scam network, which are associated with three key operations of such networks: *scamming* (Rug Pull), *wash trading* (for scam pools), and (scam) *money laundering*. All addresses must be in the network.

- S: *scammer* address - associated with a 1-day scam Rug-Pull pool (see Definition 3.2).
- C: *coordinator* address - was the *largest funder* of at least five scammer addresses, and at least 50% of its EOA neighbors must be scammer addresses. A largest funder of an address

---

[6] 0x2c1eb6ca34997f6601cffe791831ad6d5cb9e937

[7] 0x7e784333fbf355fc57de77f9fbe97aa06225acd0

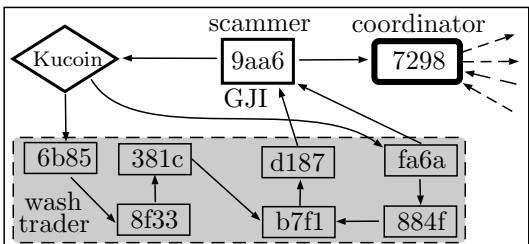

**Figure 5:** The wash traders of the scam token GJI on Uniswap created by the scammer address (ending with) **9aa6**[8] never received fund directly from it. Several are multiple hops away and funded by a CEX (Kucoin 10) or among themselves.

A is one of the in-neighbours that transferred the maximum amount of high-value token (ETH/BNB) to A.

- WT: *wash-trader* address - bought at least one 1-day scam token from a 1-day scam pool.
- D/W: *depositor/withdrawer* address - sent fund to/withdraw from CEXs, mixers, or bridges.
- T: *transfer* address - received and forwarded fund only and performed no other activities nor retained the fund received. More specifically, the address must only interact with EOAs and moreover, the total out transfers (including transaction fees) must be at least 99% of the total in transfers (not including transaction fees).
- B: *boundary* address - is not scammer or coordinator, has at least ten token swap-ins and more than 50% of the swap-ins were with non-scam pools.

## 6.2 Wash Trading and Accurate Profit Estimate

While the presence of wash traders in Rug Pulls on Uniswap was observed as early as 2021 in Xia *et al.* [38] and also noted in subsequent works [6, 15, 29], the actual cost to run wash traders for scam pools, to the best of our knowledge, has never been studied in depth. Thanks to the introduction of the *scam network* concept, we can determine the wash-trading cost more accurately.

Note that Xia *et al.* [38] identified *1-hop* wash traders that received direct transaction of ETH from scammer addresses before their swapping of scam tokens. A recently developed wash-trading detection tool by SolidusLab [18, 19] can only detect *0-hop* wash traders (self wash trading). However, from our datasets, we can see that wash traders could be multiple hops away and never interacted directly with the scammer addresses, e.g. see Fig. 5. The network of wash-trader addresses for GJI (Fig. 5) also provides an interesting *counterexample* to the common *misperception* that wash-trader addresses must receive fund from the scammer address before performing wash trading. In the example of GJI, wash traders received fund from a CEX (Kucoin) and among themselves, and even transferred the left-over amounts to the scammer address **9aa6** eventually. This would be somewhat counterintuitive if not for the knowledge of how a scam network operates.

We would also like to note that most research on wash trading for NFTs in the literature (see, e.g. [9, 22, 34] and the references therein) studied a different setting in which accounts trade NFTs

---

[8]0x19b98792e98c54F58C705CDDf74316aEc0999AA6

**Figure 6:** Part of a complete scam network containing **c6ae**, the creator of the scam token PUMPKIN. The figure shows how eight scammer/wash trader addresses transferred ETH to each other (thinner arrows) and at the same time wash traded their scam tokens (thicker arrows). The pool of PUMPKIN was heavily wash traded, thus inflating the perceived scam profit by more than 7 ETH.

*directly* among themselves or via a *centralized* marketplace like OpenSea or LooksRare, not via an exchange pool. For example, in Morgia *et al.* [22], a graph is built for each NFT with nodes being the addresses *directly* interacting with the NFT and edges being their *direct* trade of the NFT. Then, suspicious wash-trading groups include addresses that perform self-trade or strongly connected components that have a common funder or beneficiary, or maintain a zero-risk position (zero balance after all the transactions, factoring out the gas fees). The concept of zero-risk position is *irrelevant* to the exchange pool setting of fungible ERC-20/BEP-20, in which investors trade *with the pool* and *not* among themselves. Also, the edges of the connected components in the wash-trading graph in [22] represent NFT tradings, which doesn't exist in our setting as Uniswap/Pancakeswap investors do not trade ERC-20/BEP-20 tokens after buying them from the pool.

*Definition 6.1.* (Existing scam profit formula, e.g. [6, Sec. 7.1]) The profit for a scam pool $p$ is $\delta(p) \triangleq Y(p) - X(p)$, where

- $X(p)$ is the amount of high-value token (ETH/BNB) that the scammer addresses gave to the pool (by adding liquidity or swapping) *plus* transaction fees.
- $Y(p)$ is the amount of high-value token (ETH/BNB) that the scammer addresses gained from the pool (by removing liquidity or swapping) *minus* transaction fees.

For example, the creator[9] of the scam token PUMPKIN on Uniswap added 1.8 ETH (0.014 fee), swapped in twice totaled 0.22 ETH (0.0006 fee) and removed 9.27 ETH (0.0011 fee) from the pool. Ignoring the very small transaction fees, $X = 1.8 + 0.22 = 2.02$, and $Y = 9.27$, leading to a sizable profit of $Y - X = 9.27 - 2.02 = 7.25$ (more than US $11,000 back then) for Pumpkin's creator in less than an hour. We will see next that this estimation is far off from the real profit.

*Definition 6.2 (Network-aware profit formula).* The profit for a scam pool $p$ *within a scam network $N$ is $\delta(p, N) \triangleq Y(p, N) - X(p, N)$, where

---

[9]0x6C1e7FfAe984b5644C2ab95FC3aDF5794317C6aE

- $X(p, N)$ is the amount of high-value token (ETH/BNB) that the addresses in the network $N$ *paid to the pool*, including liquidity additions and swap-ins, *plus* transaction fees.
- $Y(p, N)$ is the amount of high-value token (ETH/BNB) that the addresses in the network *gained from the pool* (by removing liquidity or swapping out), *minus* transaction fees.

Note that $X(p, N)$ includes the term $X(p)$ in Definition 6.1 and the new wash-trading component $Z(p, N)$ - the total amount of high-value token paid to the pool by the wash traders in $N$.

After building the scam network containing Pumpkin's creator address using our ScamNetworkExplorer algorithm developed in Section 6.3, it becomes clear that all big investors were wash traders from that network. In particular, **4b10** and **25a1** swapped in (multiple times) 3.56 and 3.54 ETH in total, respectively, while the creator **c6ae** swapped in (twice) 0.22 ETH (see Fig. 6). Using our new formula (Definition 6.2), we obtain that $X \approx 9.1463 = 1.8177 + 7.3286$, $Y = 9.2661$, and hence the *real profit* is merely 0.1198 ETH.

Finally, the *total profit of a scam network* $N$ is simply the sum of profits of all scam pools in the network $\Delta(N) \triangleq \sum_{p \in N} \left( Y(p, N) - X(p, N) \right) - T(N)$, where $T(N)$ is the total fee spent on direct high-value token (ETH/BNB) transactions among the addresses in $N$.

## 6.3 Generating Scam Network

A natural choice to explore the scam network is to use a Breadth-First Search (BFS) as it can capture essential chain activities including fund transfer/deposit/withdraw and DEX-specific activities such as token swap and transfer, token/pool creation, liquidity providing and removal. Although theoretically straightforward, implementing BFS to identify scam networks on Ethereum and BSC chains is remarkably difficult. There are four main challenges (see Appendix F for a more detailed discussion), but the major one is to avoid network explosion and to prevent the BFS from including public or benign addresses. We will use accounts, addresses, and nodes interchangeably.

**Existing approaches.** To circumvent the network explosion problem (Challenge 4) when building the transaction subgraph (starting from one or more seed nodes), one must identify/define a boundary, or *terminal* nodes, at which the BFS stops expanding further. The simplest way is to set the boundary nodes to be just *1-hop neighbors* of the scammer nodes (equivalently, from the scam pools). For example, Morgia *et al.* [22] investigated weakly connected components of the subgraph generated by the 1-hop neighbours of the NFT scams to identify wash-trading groups. The issue with this approach is that it treats each scam as an isolated one and fails to recognize the connection among multiple scams (the main topic of our work). Another simple way is to explicitly set the *maximum number of hops* the BFS can reach, e.g. ten hops as in Yan *et al.* [39, Algo. 1]. However, this artificial threshold (ten hops) would lead to an inaccurate picture of a true scam clusters. For instance, we discovered in our work a number of very long scam chains with lengths up to a few hundreds (see Section 4.2). Limiting the BFS to a fixed, small number of hops would also lead to inaccurate statistics on scam clusters and their true profits.

**Our modified BFS** (see Appendix F) receives as input the scammer list $S$, the scam pool list $P$, and scam token list $T$ identified in the data collection phase. It then iterates over $S$, starts from each

unvisited address as a seed, retrieves the transaction history of the current address in consideration, and identifies and places its *valid* neighbors into BFS's queue for future processing. Valid neighbors of the current address $v$ include unvisited/unqueued EOAs that had a *non-zero-value transaction* (in ETH/BNB) with $v$ or were scammer addresses behind the same scam pool as $v$ (if $v$ is a scammer), were *scammer* associated with a *scam pool* that $v$ traded with, or traded with the scam pool associated with $v$ given that $v$ is a scammer.

**To address network explosion** (Challenge 4), instead of setting the maximum number of hops like in [22, 39], we allow BFS to expand arbitrarily far, but identifying *terminal* nodes at which BFS stops expanding. The set of terminate nodes is described as follows. *Public terminal nodes* are publicly label nodes such as mixers, CEXs, bridges, MEVs, contract deployers, and DEX routers. Similar to most work in the literature, a list of such addresses can be collected from well-known sites such as Etherscan's WordCloud and Dune, and also manually added on the fly. *Normal trading (boundary) nodes* are non-scammer/coordinator addresses that had at least 10 swap-ins and at least 50% of them were with non-scam exchange pool. If such an address is reached, BFS won't add its neighbors to the queue. *Big nodes* are defined according to two limits $\ell = 500$ and $L = 1000$. Addresses that have more than $L$ transaction will be ignored. Addresses with more than $\ell$ but at most $L$ transactions will be ignored except for scammers and coordinators. We also implemented an *address-poisoning-attack detector* (see Appendix F) to prevent the BFS to branch out to benign victim addresses.

**A case study.** Starting from a scam cluster found in Section 5, referred to as Cluster 126, our algorithm outputs an entire network $N$ of 240 nodes (available at [1]), including 201 scammers, 234 wash traders, three transferrers, two depositors, and two withdrawers. The total *pool profits* (across all pools) is 3.2, and after deducting the total transfer fee $T = 0.1$, the net profit is $\Delta(N) = 3.1$ ETH.

## 7 CONCLUSION

In this work, we aim to explore and understand how Rug-Pull scammers *really* worked behind devastating scams on Uniswap and Pancakeswap. We detected many scam patterns, and examined the Rug-Pull scam operations on DEX as a large-scale coordinated network rather than as isolated incidents. We found that building an entire scam network is a challenging but highly rewarding goal. With the knowledge of such a network, several interesting facts were revealed, including a) serial scammers do exist and tend to clone their scam token contracts to organize a series of scams using a large number of addresses, b) the wash traders may not have a direct contact with scammer addresses, and c) the scam profit could have been inaccurately reported due to the neglection of the significant wash-trading amounts. The introduction of scam networks allows us to unite and explain in depth several observations in the literature on how Rug-Pull scammers on DEX work. There are plenty of rooms for future research, and one of the major ones is to develop *efficient* and *accurate* network construction and address labelling algorithms that can deal with medium- and large-scale scam networks.

All important output data used in the paper is available at [1]. The Github repo of this project will be made publicly available if the paper gets accepted.

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

## A    FUNCTIONS AND EVENTS OF ERC/BEP-20

## B    ONE-DAY RUG PULL DETECTION

---

**Procedure 1** is_one_day_rug_pull(pool_address)

---

1:  lv_token ← **get_low_value_token**(pool_address)
2:  hv_token ← **get_high_value_token**(pool_address)
3:  num_pairs ← **count_pairs**(lv_token)
4:  **if** hv_token ∉ [ETH, BNB] **or** num_pairs ≠ 1 **then**
5:      **return** false
6:  mints, burns ← **get_events**(pool_address)
7:  **if** mints.length ≠ 1 **or** burns.length ≠ 1 **then**
8:      **return** false
9:  trading_time ← burns[0].timestamp − mints[0].timestamp
10: **if** trading_time > ONE_DAY **then**
11:     **return** false
12: burn_amt ← burns[0].amount
13: mint_amt ← mints[0].amount
14: burn_rate ← burn_amt/mint_amt
15: **if** burn_rate > 0.99 **then**
16:     **return** true
17: **return** false

---

**Table 5: Functions and events of the ERC-20 (Ethereum) and BEP-20 (BNB Smart Chain) standard. Among required functions, `transfer()` and `transferFrom()` are two basic functions for digital assets transferring.**

| Type | Signature | Description |
|---|---|---|
| Method | name() | Getting name of the token (e.g., Dogecoin) |
| | symbol() | Get symbol of the token (e.g., DOGE) |
| | decimals() | Get the number of decimals the token uses |
| | totalSupply() | Get the total amount of the token in circulation |
| | balanceOf() | Get the amount of token owned by given address |
| | **transfer()** | Transfer amount of tokens to given address from message caller |
| | **transferFrom()** | Transfer amount of tokens between two given accounts |
| | approve() | Allow a *spender* spend token on behalf of *owner* |
| | allowance() | Get amount that the *spender* will be allowed to spend |
| Event | Transfer() | Trigger when tokens are transferred, including zero value transfers. |
| | Approval() | Trigger on any successful call to *approve()* |

## C    DETECTING SCAM STARS

Given a list of scammer addresses, we developed `StarDetector`, an algorithm that detects all scam stars containing such addresses. First, for each scammer address $s$, `StarDetector` identifies its *major funder* as the *in-neighbour* $f$ that satisfies the following conditions: F1) $f$ funded at least 100% of the cost of the first scam carried out by $s$ with a single in-transaction before the first scam, and F2) $f$ must be the (strictly) largest funder of $s$. Similarly, `StarDetector` identifies the *major beneficiary* of $s$ as the *out-neighbour* $b$ that satisfies the following conditions: B1) $s$ transferred at least 90% of its last scam's revenue to $b$ in a single out-transaction after the last scam, and B2) the corresponding out-transaction must be the largest out-transaction after the last scam is conducted.

Next, `StarDetector` identifies the star type(s) that $s$ potentially belongs to as follows: if $f \equiv b$ then $s$ may belong to a single IN/OUT-star with center $f$; otherwise, if $f$ never received any fund from $s$ then $s$ may belong to an OUT-star with center $f$, and if $b$ never funded $s$ then $s$ may belong to an IN-star with center $b$. Note that it is possible that $s$ belongs to both an IN-star and an OUT-star (with different centers). For each star type and the corresponding potential center $c = f$ or $c = b$ or $c = f \equiv b$, the algorithm then examines each of $c$'s neighbors and checks whether $c$ is also their potential star center of the corresponding type. If there are $n \geq 5$ neighbors of $c$ (including the original scammer address $s$), then `StarDetector` returns the corresponding star, and then repeats with other addresses. The algorithm also keeps track of the star type each address already belongs to for avoiding redundant work.

## D    DETECTING MAJORITY-FLOW CLUSTERS

Finding all (maximal) majority-flow clusters among a given list $S$ of scammer addresses is a nontrivial task due to the strong properties required by the clusters. In particular, (P3) required that for every scammer address $s \in C$, its major funders either all lie inside $C$ or all lie outside $C$. Similar requirement applies to its major beneficiaries. Especially, (P4) requires that for every two scammer addresses $s$ and $s'$ in $C$, $s$ is a major funder of $s'$ *if and only if* $s'$ is a major beneficiary of $s$, which is a very strong condition, linking the funder-beneficiary addresses in *both* directions. We leave the details of the detection algorithm and its proof to Appendix D.

---

[11]The full address is 0x0bab16dff48add3db977d279f672916ddae0**9ac2**.

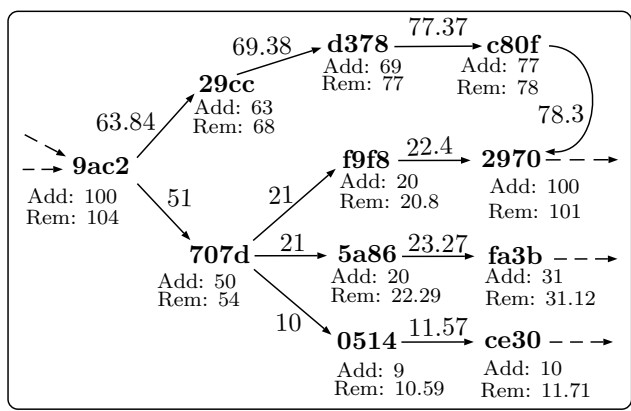

**Figure 8:** Part of a real majority-flow cluster. The address 9ac2[11]added 100BNB into a scam pool and removed 104 BNB. It then transferred 51 BNB and 63.84 BNB to its two *major beneficiaries*, 707d and 29cc, respectively. These two *internal* addresses performed one scam each and continued transferring fund to other internal addresses. Note that 2970 had two *major funders*, namely c80f and f9f8. Importantly, each internal node was *fully funded* by its major funders and also transferred at least 90% of its scam revenue (what it removed from the last pool) to its major beneficiaries.

Theorem D.1 (Majority-Flow Clusters). *Given a set $S$ of scammer addresses, each address belongs to at most one majority-flow cluster, and moreover, `FundingClusterDetector` correctly returns all maximal majority-flow clusters within $S$ (as in Definition 4.3) in time $O(|S| + |T(S)|)$, where $T(S)$ denotes the set of all transactions among scammer addresses within $S$.*

Proof of Theorem D.1. The proof is included in our supplementary materials [1]. □

We proposed `FundingClusterDetector`, an efficient algorithm that identifies all (maximal) majority-flow clusters among all scammers in $S$ in polynomial time in $|S|$. We describe below its main steps. The key idea is to first identify all *minimal* majority-flow clusters, i.e., those that satisfy (P1)-(P4), which contain no proper majority-flow sub-clusters. These minimal clusters will be the building blocks of the desired maximal clusters. This is done in Step 3,

which groups together funder-to-beneficiary transactions within $S$ that must belong to the same majority-flow clusters. Step 4 constructs the maximal clusters by connecting the transaction groups (and hence the corresponding scammer addresses) that have at least one scammer address in common. This can be done by applying a breadth-first search algorithm to the *transaction-clustering graph* introduced in Step 4.

**Step 1.** The algorithm first constructs the sets of major funders $F(s)$ and major beneficiaries $B(s)$ (if any) for every scammer address $s \in S$. Remove all $s$ from $S$ where both $F(s)$ and $B(s)$ do not exist.

**Step 2.** From the sets $F(s)$ and $B(s)$ obtained in Step 1, the algorithm builds a set of majority-flow transactions $\mathcal{T} = \cup_{s \in S}(T_F(s) \cup T_B(s))$, where $T_F(s)$ and $T_B(s)$ are the set of transactions from the major funders of $s$ to $s$ and from $s$ to its major beneficiaries (if any).

**Step 3.** The algorithm partitions the set of transactions $\mathcal{T}$ into non-overlapping groups $T_1, T_2, \ldots, T_m$ satisfying the following property: if $T_i$, $i = 1, 2, \ldots, m$, contains a transaction $t = (u, v)$ (with $u$ and $v$ being the sender and the receiver, respectively), then it must also contain all transactions $t' \in T_F(u) \cup T_B(v)$. Moreover, let $A_i \triangleq \cup_{t=(u,v) \in T_i}\{u, v\}$ be the set of all the scammer addresses being the sender or receiver in a transaction in $T_i$. This step can be implemented efficiently using a FIFO queue.

**Step 4.** The algorithm first builds the so-called *transaction-partitioning graph* $G = (V, E)$, with its vertex set $V \triangleq \{T_1, T_2, \ldots, T_m\}$ consists of all the transaction groups, and its edge set defined as $E \triangleq \{(T_i, T_j) : i \neq j, A_i \cap A_j \neq \varnothing\}$. The algorithm then finds all connected components of $G$ (e.g. by using the breadth-first search) and returns them as maximal majority-flow clusters in $S$.

## E  CONTRACT SIMILARITY

We discuss in this appendix the preprocessing step and the construction of our newly developed AST-Jaccard similiarity score that can bypass a number of obfuscation techniques when comparing scam contracts.

Popular string metrics used for measuring the source code similarity between two contracts include Levenshtein (edit) distance [20], longest common subsequences [27], and Damerau-Levenshtein distance [4]. However, such metrics are not only expensive to compute over a large number of contract pairs but also susceptible to simple code obfuscation techniques. For example, to reduce the similarity score between two contract clones, scammers can employ techniques like (M1) comments and spaces insertion/deletion, (M2) identifier/variable names modification, or (M3) code reordering to increase the edit distance arbitrarily. To tackle these, we propose a token-based similarity score call AST-Jaccard score, which leverages source code's AST, a collision-resistant hash function, and Jaccard similarity to compare scam contracts.

More specifically, as AST keeps the syntax and sematic information of the source code while ignoring non-essential information such as spaces, comments, specific function and variable names, using AST instead of the source code itself tackles (M1) and (M2). To bypass (M3) code reordering, e.g. swapping functions or variables around, the new score uses the Keccak-256 hashing algorithm to hash all tokens/components in the AST of each source code and group them into a set. As the order of elements is not important for a set, even when the variables and function were rearranged,

the Jaccard similarity on sets still pick up the right overlapping ratio between the two sets corresponding to two contract codes. To further improve the accuracy, we also applied a preprocessing steps to remove all common libraries and interfaces in every source code, which account for 40%-50% of the code itself and would interfere with the score. More details about the preprocessing and the AST-Jaccard score can be found in Appendix E.

**Common Libraries and Interfaces Removal.** Fungible tokens are implemented by following the common standard (e.g ERC-20 or BEP-20 interface) so these tokens often contains the common codes from the interface, inflating their similarity. The similarity continues to increase if they used the same common libraries such as `SafeMath`, `Address`, or `Ownable`. Hence, to avoid this inflation, we remove all common libraries and interfaces from tokens before measuring their similarities. To that end, we first collect all common libraries on OpenZeppelin [25], an open-source framework for writing secure and scalable smart contracts. Then we remove all these libraries and interfaces from token's contracts before doing tokenisations.

**Code Tokenisation.** Our approach first parses the source code of a contract to an AST, which keeps the syntax and semantic information of a contract only. Subsequently, we extract a token from a type of each node in the AST (e.g., `Mapping`, `IfStatement`, `BinaryOperation`, `Assignment`). In this manner, some inessential information such as spaces, comments, function names, variable names and their values will be eliminated. Thus, we can solve M1 and M2. We run an extraction for each component in a contract, such as *state variables*, *functions*, *events*, and *modifiers*. Notably, code lines in a component will be parsed to an array of tokens. For example, we can get correspondingly an array of *IfStatement*, *BlockIdentifier*, *IndexAccess*, *BinaryOperation*, and *Literal* tokens from the `if` condition "if(sender[i] == '0x0d83a1')". In other words, each array of tokens marks the semantic information of each component in a contract.

**Integration of AST, hash function, and Jaccard similarity Score.** Next, we try to overcome M3 at a contract level, i.e. to identify a clone of a contract even if the scammer has reordered different components in the contract. Our approach does not focus on a component level because we are unsure whether different code line arrangements in a component will give out the same working logic or not (e.g., a code line is inside or outside a condition block) while our main goal is to build a ground truth dataset for a Trapdoor detection problem. Therefore, two contracts are totally the same if they contain the same list of components. To improve our algorithm's performance, we concatenate all tokens in the token array of a component as a string and apply a Keccak-256 hash algorithm. As a result, each component in the contract will be represented as a unique hash (256 bits). Next, we employ the classic Jaccard index [37] for two corresponding sets of hashes as in (1), where $J(A, B)$ is the similarity between two contracts, $A$ is a list of hashes of the first contract and $B$ is a list of hashes of the second contract.

$$J(A, B) = \frac{|A \cap B|}{|A \cup B|} = \frac{|A \cap B|}{|A| + |B| - |A \cap B|}. \tag{1}$$

The overall algorithm is presented below.

---

**Procedure 2 tokenization**(token)

1: ast ← **parseAST**(token.contract)
2: components ←**removeCommon**(ast.contract.nodes)
3: hashes ← []
4: **for** j ← 0 to components.length **do**
5:   syntactic_tonkens ← **extractTokens**(components[i])
6:   token_str ← **concat**(syn_tonkens)
7:   hashes[i] ← **keccak256**(token_str)
8: **return**  hashes

---

**Procedure 3 similarity**(tokenA, tokenB)

1: hashesA ←**tokenization**(tokenA)
2: hashesB ←**tokenization**(tokenB)
3: intersection ← hashesA & hashesB
4: union ← hashesA | hashesB
5: jaccard_index ← intersection.length / union.length
6: **return**  jaccard_index

---

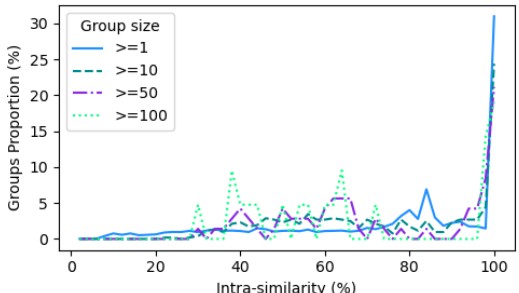

**Figure 9:** *Intra-cluster* similarities on Uniswap.

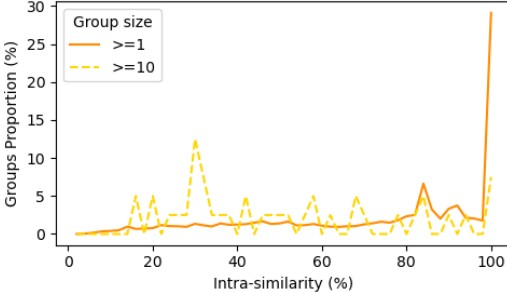

**Figure 10:** *Intra-cluster* similarities on Pancakeswap.

## F   SCAM NETWORK CONSTRUCTION

There are many challenges when constructing a scam network using BFS.

**Challenge 1.** *The background graph is non-homogeneous.* Indeed, nodes in the transaction graph include externally owned accounts (EOAs) and contract accounts. Contract accounts could be a mixer, a centralized exchange, a bridge, a router, a bot (MEV, trading bots), a token deployer, each type behaves differently and requires a different treatment.

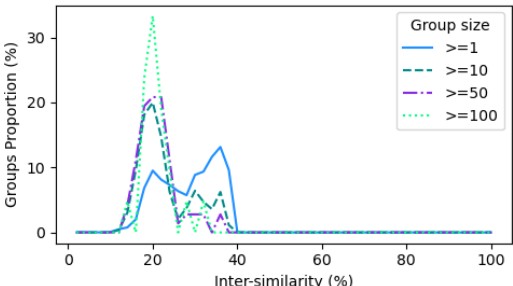

**Figure 11:** *Inter-cluster* similarities distribution on Uniswap.

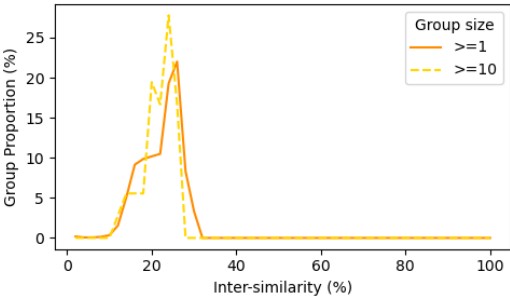

**Figure 12:** *Inter-cluster* similarities on Pancakeswap

**Challenge 2.** *Identifying an edge between two accounts is challenging and requires domain knowledge.* For example, an edge exists between two EOAs A and B not only when A transferred fund to B directly, but also when A buys a scam token created by B or invested in a scam pool rugged by B. On the other hand, if A sends a 0-value transaction to B (a message-carrying transaction), that shouldn't be counted as an edge[12].

**Challenge 3.** *The background graph is huge and unavailable.* By contrast to the standard setting in the traditional graph theory, due to its sheer size, the complete underlying transaction graph among all accounts on Ethereum or BSC (or any other established chain) is not available and impossible to build[13].

**Challenge 4.** *Network explosion.* Starting from a (seed) scammer node, the standard BFS can quickly expand to include an unmanageable number of nodes in its queue, especially when the node in consideration starts transacting with public accounts (CEXs, mixers, bridges), or trading from non-scam pools. Thus, we must be able to recognize the boundary nodes associated with non-scam activities, to prevent BFS from including normal nodes unrelated to the scammers, which may have been active for years with thousands to hundreds of thousands of transactions and would fairly quickly overwhelm BFS's memory and pollute the real scam cluster.

Apart from the above, irrelevant transactions generated by *phishing attacks* (e.g. address-poisoning attacks) can also confuse the BFS, making it jump out of the scam network by mistake.

---

[12]Our algorithm initially hit a non-scam account that sent a transaction to a known scam account. It turns out that the transaction only carries a sneering message.
[13]For example, the Ethereum chain generates more than a million new transactions everyday (see https://ycharts.com/indicators/ethereum_transactions_per_day).

**Procedure 7 get_labels**(address, normal_txs, internal_txs)

1: labels ← []
2: token_buys ← **count_swap_in**(normal_txs)
3: scam_token_buys ← **count_scam_swap_in**(normal_txs)
4: scam_buy_rate ← token_buys/scam_token_buys
5: valid_EOA ← **get_valid_neighbours**(address, normal_txs)
6: funded_scammers ← **get_funded_scammers**(valid_EOA)
7: scammer_rate ← funded_scammers/valid_EOA
8: fo_rate ← **get_fund_out_rate**(normal_txs)
9: **if** address ∈ scammers **then**
10:     labels.**append**(S)
11: **if** funded_scammers.length > 5 **and** scammer_rate > 0.5 **then**
12:     labels.**append**(C)
13: **if exist_tx_to_exchange**(normal_txs) **then**
14:     labels.**append**(D)
15: **if exist_tx_from_exchange**(normal_txs) **then**
16:     labels.**append**(W)
17: **if** scam_token_buys.length > 0 **then**
18:     labels.**append**(WT)
19: **if** S ∉ labels **and** C ∉ labels **and** token_buys.length > 10 **and** scam_buy_rate < 0.5 **then**
20:     labels.**append**(B)
21: **if** !**exist_contract**(normal_txs) **and** fo_rate ≥ 0.99 **then**
22:     labels.**append**(T)
23: **if** normal_txs.length >= 1000 **then**
24:     labels.**append**(LIMIT_1)
25: **else if** S ∉ labels **and** C ∉ labels **and** normal_txs.length >= 500 **then**
26:     labels.**append**(LIMIT_2)
27: **return** labels

**Procedure 8 create_node**(address)

1: valid_EOAs ← []
2: normal_txs, internal_txs ← **get_transactions**(address)
3: labels ← **get_labels**(address, normal_txs, internal_txs)
4: **if** B ∉ labels **and** LIMIT_1 ∉ labels **and** LIMIT_2 ∉ labels **then**
5:     valid_EOAs ← **get_valid_neighbours**(address, normal_txs)
6: node ← Node(address, valid_EOAs, labels)
7: **return** node

**Procedure 9 explore_network**(cluster)

1: pioneer ← cluster[0]
2: node ← **create_node**(pioneer)
3: node.valid_EOAs.**extend**(cluster[1:])
4: queue ← Queue
5: network ← Network
6: traversed ← set()
7: queue.**enqueue**(node)
8: network.**add_node**(node)
9: **while** queue.length > 0 **do**
10:     root ← queue.**dequeue**()
11:     **if** root.address ∈ traversed **then**
12:         **continue**
13:     traversed.**add**(root.address)
14:     **for** i ← 0 to root.valid_EOAs.length **do**
15:         eoa ← root.valid_EOAs[i]
16:         **if** eoa ∉ traversed **and** eoa ∉ queue **and** eoa ∉ network **then**
17:             nbnode ← **create_node**(eoa)
18:             labels ← nbnode.labels
19:             **if** !**is_phishing**(root, nbnode) **and** B ∉ labels **and** LIMIT_1 ∉ labels **and** LIMIT_2 ∉ labels **then**
20:                 queue.**enqueue**(nbnode)
21:                 network.**add_node**(nbnode)
22: **return** cluster

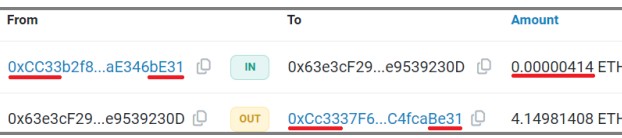

**Figure 13:** The phishing address has the same (lower-cased) first and last four digits as an out-neighbor of the victim address. It transferred a tiny amount to the address **230d**, hoping that one day the address owner will mistakenly transfer fund to it instead.

**Address-poisoning attacks can break BFS.** Phishing attackers target everyone, including addresses in Rug-Pull scam networks. Dusting attack or dust value transfer (see, e.g. [3, 10]) is a common type of address poisoning attack in which an address that looks very similar to the victim's out-neighbor (see Fig. 13) transferred a tiny amount to the victim address. A naive BFS may expand from a node in the scam network that was the target of a dusting attack to visit the attacker address, then to its coordinator address (the one who coordinates all the phishing attacks) and then unknowingly to *all* of its (benign) victim addresses in the network. We implemented a simple phishing detector that excludes a phishing address from the set of valid neighbors for each address in BFS, which detected dozens to hundreds of such addresses for every network we tried.

**Procedure 4 is_contract**(address)

1: bytecode ← web3.eth.**get_code**(address)
2: **if** bytecode.length > 0 **then**
3:     **return** true
4: **return** false

**Procedure 5 is_public**(address)

1: CEX, DEX, MEV, MIXER ← **load_public_addresses**()
2: end_nodes ← CEX ∪ DEX ∪ MEV ∪ MIXER
3: **if** address ∈ end_nodes **then**
4:     **return** true

---

**Procedure 6 get_valid_neighbours**(address, normal_txs)

---

1: valid_EOAs ← []
2: **for** i ← 0 to normal_txs.length **do**
3:    s ← normal_txs[i].from
4:    r ← normal_txs[i].to
5:    v ← normal_txs[i].value
6:    cf ← normal_txs[i].functionName
7:    **if** r = address **and** v > 0 **and** !**is_public**(s) **then**
8:      valid_EOAs.**append**(s)
9:    **else if** !**is_contract**(r) **and** v > 0 **and** !**is_public**(r) **then**
10:     valid_EOAs.**append**(r)
11:    **else if** cf ≠ NULL **then**
12:     scammers ← **get_scammers_if_swap**(ip)
13:     valid_EOAs.**extend**(scammers)
14: **return** valid_EOAs

---