# OpenReview forum: "Serial Scammers and Attack of the Clones: How Scammers Coordinate Multiple Rug Pulls on Decentralized Exchanges"
_ACM.org/TheWebConf/2025/Conference — WWW 2025 Oral_

### Official Review · Reviewer_BCLV · 2024-11-24

**Novelty:** 3
**Technical Quality:** 5

**Review:**

This paper presents a measurement method to identify the 1-day Rug Pull serial scammer in Ethereum and BSC. It reconstructs the scam networks and estimates more accurately profits of the scammers. Specifically, this paper analyzes rug-pull scams on Uniswap and Pancakeswap V2. It defines and identifies key scam patterns, including star-shaped, chain-shaped, and majority-flow clusters to show how serial scammers organize multiple scams. Using a scam network explorer algorithm, it reconstructs the operations of these scammers and introduces a network-aware profit formula that accounts for wash-trading expenses, providing a more precise estimation of scam profits.

+ This paper conducts a large-scale analysis of Uniswap and Pancakeswap, examining 356,295 pools and 343,637 tokens on Uniswap, and 1,694,058 pools and 1,510,774 tokens on Pancakeswap. It identifies 161,339 scam pools and 145,654 unique scammers on Uniswap, and 32,336 scam pools and 18,346 unique scammers on Pancakeswap.
+ Their method for reconstructing the scam network addresses a gap in existing research, enabling the identification of scam clusters and uncovering multi-hop wash trading activities.
+ Their method provides a more accurate estimation of scam profits by uncovering multi-hop wash trading patterns and accounting for the associated wash-trading expenses, offering insights into the financial profit of scams.

- Limited experimental evaluation: The paper provides a case study of scam network generation and profit estimation. Expanding the analysis to include a large-scale evaluation of scam profits could provide deeper insights into the broader economic impact of rug-pull scams.
- Absence of comparison with existing methods: The paper does not benchmark its approach against existing techniques.
- The impact and outcomes of this measurement work remain unclear. While the authors present findings such as the existence of serial scammers, the possibility of wash traders operating without direct links to scammer addresses, and inaccuracies in previous scam profit estimations, these insights feel somewhat superficial.

**Questions:**

- My biggest concern is whether there is a ground truth available to evaluate the false positives in address identification and the correctness of profit estimation in your methods.
- Can you provide the rationale behind the thresholds chosen in your method, such as "burns at least 99% of the minted LP tokens," "price dropped by more than 90%," and "transfers at least 90% of the last scam revenue"?
- Can you provide a large-scale evaluation of scam profits and provide deeper insights?
- Can you provide the evaluation results comparing your method with existing approaches?

Miner issue: Typo in P3, “s cam”

**Reviewer Confidence:**

3: The reviewer is confident but not certain that the evaluation is correct

**Scope:**

3: The work is somewhat relevant to the Web and to the track, and is of narrow interest to a sub-community

---

### Official Review · Reviewer_3hs8 · 2024-11-28

**Novelty:** 4
**Technical Quality:** 3

**Review:**

This paper uncovers potential links between different one-day rug pull incidents through fund flow analysis, suggesting that seemingly unrelated events may, in fact, be orchestrated by the same real-world entities (i.e., serial scammers).

### Pros

- The research issues mentioned in this paper, such as clustering multiple scammer addresses, play a crucial role in enhancing the understanding and analysis of rug pull incidents.
- This paper uncovers several interesting patterns in the fund flows between scammer addresses, providing valuable insights into their transactional behaviors.
- It identifies and clarifies several distinct types of addresses involved in rug pull incidents, along with their specific functions (e.g., wash trading, coordination). Building on this understanding, the paper proposes a more accurate method for estimating the profits generated from these scams.
- The four main challenges discussed in "Generating Scam Network" is spot-on, and this paper makes an effort to propose solutions to these challenges.

### Cons

- Section 4 introduces three patterns of fund interactions among scammer addresses. However, these patterns are restricted to native token transfers (ETH or BNB), and the determination of transfer amounts seems to rely heavily on empirical thresholds, such as requiring at least 90% of the scam revenue. This reliance raises concerns about the generalizability and universality of these patterns across diverse scam scenarios.
- If my understanding is correct, the "Scam Cluster" construction method proposed in Section 5 is not well-designed, as it only considers direct edges between **scammer addresses**. This approach overlooks potential connections through intermediary non-scammer addresses. For instance, consider an upstream scammer-controlled address $u$ that transfers 10 ETH to $s_1$ via the path  $u \rightarrow v_1 \rightarrow s_1 $ and also transfers 10 ETH to $ s_2 $ via the path $ u \rightarrow v_2 \rightarrow s_2 $. Both  $ s_1 $ and $ s_2 $ use these funds to execute separate rug pull scams. (Note that, $u$, $v_1$, $v_2$, $s_1$ and $s_2$ are all controlled by the same scammer, and that $u$, $v_1$, $v_2$ are not *directly* associated with any scam pools.) In such cases, there is strong evidence to group $ s_1 $ and $ s_2 $ into the same cluster. However, the method proposed in Section 5 would fail to establish this connection, as it does not account for links mediated by non-scammer addresses. This clustering method is not sufficiently robust, which may explain why many clusters (i.e., 92% Uniswap clusters and 88% Pancakeswap clusters) contain only a single scammer address.
- The technical focus of Section 6 lies in the generation of the scam network. While the paper proposes a BFS algorithm specifically tailored to this scenario, the approach lacks significant technical innovation and overlooks several real-world issues, such as the complexity of multi-token transactions, and interactions involving smart contracts that trigger fund transfers between scammer addresses.

If I have misunderstood any part of the paper, I sincerely apologize and kindly ask the authors to clarify in their response. I also have some additional concerns, which are outlined in the following Questions section.

**Questions:**

1. What proportion of all scammer addresses align with the three patterns proposed in Section 4? Do the scammer addresses that do not conform to these patterns exhibit more complex or nuanced fund interaction behaviors?
2. Can the Scam Network generation method proposed in Section 6 identify the following type of wash trading behavior and accurately estimate profits: Address $a$, controlled by the scammer, uses 1 ETH to purchase 1000 scam tokens from the scam pool. Address $a$ then transfers the 1000 scam tokens to another scammer-controlled address $b$. Subsequently, address $b$, acting as a wash trader, engages in repeated buy-and-sell transactions within the scam pool to artificially inflate trading activity?
3. The authors may address the concerns raised in the "Cons" by providing counterarguments and clarifying the facts.

**Reviewer Confidence:**

3: The reviewer is confident but not certain that the evaluation is correct

**Scope:**

4: The work is relevant to the Web and to the track, and is of broad interest to the community

---

### Official Review · Reviewer_gSpN · 2024-12-01

**Novelty:** 3
**Technical Quality:** 3

**Review:**

This paper aims to analyze the pattern of serial scammers and attacks on decentralized exchanges. From the abstract, it is clear that the overall paper looks like an engineering technique report rather than a well written scientific paper. This point comes from the fact that there is no clear primary research question in this paper. In another way, authors made great efforts to preprocess the data, build the network, figure out the patterns. Such flow can be necessary as an engineering project. However, it needs to figure out the primary research questions and studied carefully. The figures and tables have poor qualities below the bar, please read high-quality papers carefully before submit the papers.

**Questions:**

1	The figures and tables have poor qualities below the bar, please read high-quality papers carefully before submit the papers.
2.  The introduction part of this paper is lengthy, which could be simplified to focus on research questions within 3 paragraphs.
3. From the contribution parts, it is known that data preprocessing, scam patterns, grouping of scammer addresses and scam profit estimation. To my best knowledge, the first three contributions are not scientific contributions because there are no new idea or methods presented there. For the last one, it is a simple notion rather than qualified scientific contribution. Such things might be valuable for junior conferences, but it is definitely not qualified to WWW with good reputation.
4. I strongly suggested the authors carefully formulate the primary research question and revised the paper thoroughly.

**Reviewer Confidence:**

3: The reviewer is confident but not certain that the evaluation is correct

**Scope:**

3: The work is somewhat relevant to the Web and to the track, and is of narrow interest to a sub-community

---

### Official Review · Reviewer_nUvV · 2024-12-01

**Novelty:** 4
**Technical Quality:** 2

**Review:**

Strengths:

The paper provides novel insights into how serial scammers operate coordinated rug pull schemes. The authors develop new methodologies to identify and analyze scam patterns, reconstruct complete scam networks, and more accurately estimate true profits by accounting for wash trading costs. The work fills important research gaps highlighted in previous studies by demonstrating how seemingly independent scams are actually coordinated by the same actors.

Weaknesses:

1. Focus is limited to 1-day rug pulls only, excluding longer-duration scams

2. Analysis covers only ETH pools on Uniswap and BNB pools on PancakeSwap

3. Time period of data collection is not clearly specified in some places

4. These pattern detection methods primarily rely on predefined pattern types and fixed threshold parameters, focusing mainly on direct fund flow relationships while potentially overlooking more complex indirect relationships, and lacking consideration of time series features. Furthermore, the computational complexity of these methods increases with network size, potentially making them difficult to scale to larger datasets. Most importantly, as scammer strategies continue to evolve, new scam patterns may emerge that are not covered by these predefined patterns. Therefore, while the paper's methods can effectively identify some typical scam patterns, they may not be able to discover all scam patterns, indicating the need for developing more flexible and comprehensive detection methods in the future.

5. Limited discussion of how findings could be used for scam prevention; No concrete recommendations for exchanges or regulators; Mitigation strategies are not explored in depth

6. Some implementation details of key algorithms are left to appendices, and some Procedure5 is incomplete

7. Parameters choices (e.g., thresholds) could use more justification

8. Computational complexity analysis is limited

**Questions:**

1. How sensitive are your results to the chosen thresholds (e.g., 100% for funding and 90% for revenue in star patterns)? Did you experiment with different thresholds?

2. Could your pattern detection methods miss sophisticated scammers who intentionally structure their operations to avoid these patterns?

3. Why focus exclusively on 1-day rug pulls? What percentage of total rug pulls might you be missing by excluding longer-duration scams?

4. Have you considered applying your methods to other DEXs or blockchain networks beyond Uniswap and Pancakeswap?

5. How do you validate the accuracy of your scam network reconstruction? What is your false positive/negative rate?

6. The paper mentions challenges with network explosion - how do you ensure your boundary node criteria don't prematurely terminate exploration of legitimate parts of scam networks?

7. How do you validate your identification of wash traders, especially those multiple hops away from scammer addresses?

10. Could your profit estimation still be inaccurate if wash traders use multiple accounts or more sophisticated methods to hide their activities?

11. What recommendations would you make to DEX platforms based on your findings to help prevent these coordinated scams?

12. How robust is your AST-Jaccard similarity measure against more sophisticated code obfuscation techniques?

13. What threshold of similarity do you use to determine contracts are clones, and how did you validate this threshold?

**Reviewer Confidence:**

4: The reviewer is certain that the evaluation is correct and very familiar with the relevant literature

**Scope:**

3: The work is somewhat relevant to the Web and to the track, and is of narrow interest to a sub-community

---

### Official Review · Reviewer_KCbW · 2024-12-02

**Novelty:** 5
**Technical Quality:** 4

**Review:**

Description

The paper presents a study on decentralized exchanges for identifying and
characterizing Rug Pulls. In cryptocurrency applications, a Rug Pull is a scam,
where a new project/token is advertised and then it is suddenly discontinued
with developers (which act as scammers) disappearing with all the funds.


Pros

- The datasets look representative.

- Some conclusions from the analysis are new.

Cons

- Unclear positioning in the related work.

- Some events characterized as Rug Pulls in the study may not be actual scams.

- Exploring how to use such results for detection is missing.

- Weak connection with WWW research.

Details

- Overview. The paper presents a study of Rug Pulls in decentralized exchanges.
  Although I am not familiar with this area, according to the authors, such
  study, where multiple addresses operate on multiple scam pools, has not been
  carried out in the past. Having said that, a related-work section, with
  comparison of similar efforts and positioning to the current state of the art
  would be certainly beneficial for the reader. Towards this, the dataset and
  the analysis seem thorough and some new conclusions derived by this work may
  be beneficial for the community.

- False positives. It is uncertain if all events that are captured by the
  definition of 3.1 are actually scams and not other events that are not on
  purpose malicious. In other words, I am not entirely sure if the definition
  of 1-Day Rug Pull is strict enough to capture only malicious events.

- Deploying defenses from the analysis. The paper does not explore how someone
  can realize mitigations or detecting mechanisms for Rug Pull events, based on
  this study. It would be interesting to show, how decentralized exchanges
  could investigate such campaigns or build systems that can detect Rug Pulls
  early.

- Connection to WWW. The concepts presented in this paper are not directly
  related to research of WWW, but are mostly connected with studying of scams
  in cryptocurrencies.

**Questions:**

- Can you be certain that you have no false positives in your analysis?

- How can someone deploy defenses based on your conclusions?

**Reviewer Confidence:**

1: The reviewer's evaluation is an educated guess

**Scope:**

1: The work is irrelevant to the Web